# Peripheral Intravenous Therapy in Internal Medicine Department—Antibiotics and Other Drugs’ Consumption and Characteristics of Vascular Access Devices in 2-Year Observation Study

**DOI:** 10.3390/antibiotics13070664

**Published:** 2024-07-18

**Authors:** Piotr Piekiełko, Anna Mucha, Ewa Stawowczyk, Jadwiga Wójkowska-Mach

**Affiliations:** 1Department of Internal Diseases and Circulatory Failure, Center of Pulmonology and Thoracic Surgery in Bystra, Juliana Fałata 2 Street, 43-360 Bystra, Poland; ppiekielko@szpitalbystra.pl; 2Department of Pulmonology and Respiratory Failure, Center of Pulmonology and Thoracic Surgery in Bystra, Juliana Fałata 2 Street, 43-360 Bystra, Poland; 3Department of Pulmonology and Oncology with Chemotherapy, Center of Pulmonology and Thoracic Surgery in Bystra, Juliana Fałata 2 Street, 43-360 Bystra, Poland; amucha@szpitalbystra.pl; 4Department of Microbiology, Faculty of Medicine, Jagiellonian University Medical College, Czysta 18 Street, 31-121 Krakow, Poland; estawowczyk@gmail.com

**Keywords:** antibiotic treatment, vascular access device, peripheral intravenous catheter, intravenous therapy, internal medicine, geriatric patients

## Abstract

Background: The aim of the study was to characterize the procedure of peripheral intravenous therapy (IT), including the characteristics of vascular access and related complications and qualitative and quantitative analyses of drug consumption. Materials and Methods: A two-year, retrospective, single-center observational study was conducted. The criterion for including a patient in the study was the use of peripheral intravenous catheters (PIVCs) upon admission or during the stay at the internal medicine department (IMD). Results: The main reasons for hospitalization were exacerbations of chronic diseases for 78% of the patients and acute infections for 22%. IT was used in 83.6% of all the patients. IT was used primarily for antibiotics (5009.9 defined daily doses (DDD)). Further, 22.6% of the PIVCs stopped functioning within 24 h, more frequently in infectious patients. The main reasons for PIVC removal were leakage (n = 880, 26.6%) and occlusion (n = 578, 17.5%). The PIVC locations were mostly suboptimal (n = 2010, 59.5%), and such locations were related to leakage and occlusion (*p* = 0.017). Conclusions: In the IMD, most patients require the use of a PIVC, and antibiotics dominate the group of drugs administered intravenously. Up to 1/5 of peripheral intravenous catheters are lost within the first 24 h after their insertion, with most of them placed suboptimally. A properly functioning PIVC appears to be crucial for antimicrobial treatment.

## 1. Introduction

The intravenous administration of drugs, fluids, blood products, and nutrition is one of the most common procedures in acute hospitals, affecting most of the patients [1]. Many acute conditions and exacerbations of chronic diseases significantly deteriorate patients’ well-being and exclude the possibility of full treatment with oral preparations [2]. On the other hand, many drugs are only available parenterally, especially broad-spectrum antibiotics, which must be delivered immediately in life-threatening infections [3]. The entire intravenous therapy (IT) procedure includes vascular access, its maintenance, and all the substances administered during the procedure. A vascular access device (VAD) is an essential component in IT as it enables the entrance into a patient’s blood vessel system. An appropriate VAD enables the delivery of the planned therapy, has a low risk of complications, and prevents the need for further procedures by protecting the vessels. The most common type of VAD is a peripheral intravenous catheter (PIVC) due to its minimally invasive and relatively easy insertion technique. The European recommendations state that a PIVC should be used for an anticipated IT duration of 2 to 7 days and in emergencies [4]. In a medical treatment team, the nursing staff are usually the IT supervisors. The success of IT using a PIVC is dependent on many factors related to the patient’s clinical status and behavior and the nursing staff decisions. It is the responsibility of the nursing staff to choose the appropriate place for the VAD insertion and to take care of the vascular line. The patient’s responsibility is to be aware of having a catheter, its limitations, and reporting disturbing symptoms that may indicate its malfunction [5]. Nevertheless, there is a growing awareness of the imperfections of this technique. PIVCs have a high risk of failure and may lead to complications such as phlebitis, occlusion, infiltration, dislodgement, and local or systemic infection. The treatment can rarely be completed with a single catheter, leading to a vicious process of subsequent insertions; this not only causes unpleasant sensations for the patient and the loss of more peripheral vessels but also increases the risk of infection [6,7]. Antibiotics, despite their undeniable role in treating infections, are also associated with increased PIVC failure through their potentially irritating effects on blood vessels for some of them, such as cloxacillin, vancomycin, ceftazidime, and cefepime [8]. The loss of a VAD may also delay treatment, deteriorating the patient’s condition and prolonging hospital stays. Using IT seems to be one of the keys to ensuring proper care, especially in the internal medicine department (IMD), which provides a wide range of medical services and comprehensive care in the treatment of non-surgical diseases, especially for older patients [9]. Therefore, proper supervision during catheter insertion and while maintaining the vascular line during hospitalization is even more important because it can reduce the occurrence of phlebitis and bloodstream infections [10]. Providing VAD care and maintaining proper supervision during IT remain great challenges; this was especially true during the COVID-19 pandemic, substantially increasing the number of patients requiring care [11,12].

Our study aimed to characterize the procedure of implementing peripheral intravenous therapy, including vascular access and related complications and considering the qualitative and quantitative characteristics of the drugs most frequently used by the IMD for this therapy, during a two-year observation period.

## 2. Results

During the analysis period, 1406 patients were hospitalized, and the eligibility criteria were met by 1176 patients, constituting 83.6% of all the hospitalized patients. The total PIVC patient days were 9678, and the PIVC utilization ratio was 0.8. Most of the patients with a PIVC were admitted due to an emergency, specifically 1029 patients or 87.4%. The main reason for hospitalization was an exacerbation of a chronic disease, especially cardiovascular, gastrointestinal, and metabolic disorders; 22% of the patients were treated due to infectious diseases, with 73 cases of laboratory-confirmed BSI, which accounted for as much as 28% of all the infections. A total of 159 patients were admitted due to COVID-19 infections; urinary tract infections and enterocolitis constituted 20 cases and other infections constituted 19 cases. One in five patients had a previous hospitalization up to three months prior. The median (Me) age was 74 years (interquartile range (IQR) 66–83). The Me LOS was 9 days (IQR 6–12), and the Me duration of IT was 7 days (IQR 5–11) (Table 1). The number of deaths in the entire population hospitalized during the studied period was 158, while, among the patients with PIVC, it was 130. The PIVC fatality case rate was 11.1%, similar compared to the total fatality case rate, which was 11.2% (Table 1). 

A total of 3377 PIVCs were analyzed, with the Me PIVCs’ number equal to 2 (IQR 1–4). In 36.1% (n = 425) of the patients, at least one of the inserted cannulas stopped functioning within 24 h of insertion, including 764 catheters (22.6% of all), with a dwell time below one day. The patients with catheters that stopped functioning within 24 h were older (75 vs. 71 years, *p* < 0.001), the LOSs were longer (12 vs. 9 days, *p* < 0.001), and the IT durations were longer (10 vs. 7 days, *p* < 0.001) than those of the patients without such events within 24 h (Table 2). The patients hospitalized due to infections were more likely to have at least one PIVC that functioned for <24 h, respectively, 52.1% vs. 47.9%, *p* < 0.001 (Figure 1). 

The most common PIVC locations were the forearm (28.3%), the hand (23.3%), and the antecubital fossa (17.7%). The main reasons for cannula removal were the discontinuation of therapy (31.1%), leakage (26.6%), and obstruction (17.5%) (Figure 2). 

A correlation was found between the location of the PIVC in a suboptimal place and the occurrence of complications, such as occlusion and leakage (*p* = 0.017). A total of 242 (7.3%) catheters had mild signs of venous inflammation at the time of removal (Table 2). During the study period, 16 central venous catheters were placed, all because PIVC placement was not possible.

The main types of IV therapy were antimicrobials with 5179.2 DDDs, 33.4% of all the PIVC days, and glucocorticosteroids with 3746.7 DDDs, 24.2% of all the PIVC days. One of the largest groups of substances was infusion fluids, with 5,705,600 mL (Table 3). 

In total, 6410 DDDs of antimicrobials were used, and the total oral and intravenous consumption of antibacterials for systemic use was 600 DDDs per 1000 patient days. Intravenous administration was used more compared to oral, with 5009.9 DDDs and 1400.1 DDDs, respectively (78.2% vs. 21.8%). The most common antimicrobials used in IT were beta-lactams, accounting for 3995.5 DDDs (Table 4). In 2022, compared to 2021, there was an increase in the consumption of IV beta-lactams (2256 DDD vs. 1739.6 DDD) and aminoglycosides (84.8 DDD vs. 36.5 DDD), and a decrease in the consumption of imidazole derivatives (67 DDD vs. 159.7 DDD). In oral administration, increases in penicillin (127 DDD vs. 62.3 DDD), sulfonamides, and trimethoprim (272 DDD vs. 160 DDD) were noted, as well as decreases in the other antibiotics from the beta-lactam group (21 DDD vs. 90 DDD) (Table 4). The antibiotic agents likely to be associated with catheter failure among all the intravenous antibiotics constituted 21.8% (1091.5 DDD). Cloxacillin was the majority, with 753.5 DDD, followed by vancomycin with 325.5 DDD. The ceftazidime consumption was 12.5 DDD, and cefepime was not used (Figure 3). 

## 3. Discussion

In the studied department, the patients with PIVC constituted 83.6% of all those hospitalized, and up to 1/5 of the peripheral catheters stopped functioning before 24 h. The length of the IT therapy was 7 days compared to 9 days for hospital stay length. Most of the PIVCs were located in suboptimal places. In terms of peripheral intravenous therapy, antibiotics were the dominant group of administered drugs. Furthermore, regarding the total antibiotics consumption, the IV route was the main one.

The patients with PIVCs constituted the vast majority of those hospitalized; therefore, all the procedures related to VADs and IT should be prioritized in the described department. A notable and very disturbing fact is the relatively large number of patients in whom at least one PIVC stopped functioning before 24 h due to pathological reasons (36.1%), with the total number of these catheters being over 20%. According to the recommendations, a single PIVC should provide vascular access for approximately 7 days of IT, but the results indicate the need to use as many as two peripheral catheters for a treatment of this duration [4,13]. The geriatric population is characterized by an atypical symptomatology of infectious diseases. Acute disturbances of consciousness, like delirium or confusion, can often be observed as the first signs of an infection. Decreased cognitive function prevents proper communication between the medical staff and the patient, which may result in a lack of cooperation during treatment, especially when using vascular access devices [14]. This is probably why advanced age and more frequent infections were the distinguishing features of the patients who lost catheters within 24 h of insertion. The use of antibiotics related to the development of complications such as vasculitis accounted for as much as 1/5 of the total consumption, which could also have a significant impact on the results [8].

The Vessel Health and Preservations initiative [15] emphasizes numerous issues regarding the proper construction and management of the vascular line to reduce the number of cannulations and associated loss of vessels necessary for future procedures. One of the most important issues is selecting the optimal vascular access site [16]. In accordance with the guidelines, the best site for PIVC insertion is the forearm due to its low mobility, which increases the probability of maintaining the cannula throughout the therapy period, protects it against accidental removal, and reduces negative feelings during infusion [17]. In our study, the PIVCs were placed in the forearm in only 28% of the cases. More than half of all the peripheral cannulas (59.5%) were sited in non-optimal places, such as the hand, antecubital fossa, wrist, and foot. These constitute areas of flexion, which should be avoided due to the higher risk of complications. The lower extremities are especially excluded from access due to the higher risk of thrombophlebitis and ulceration in adults. PIVC insertion in the hand can be considered only for short-term therapy lasting less than 24 h. The antecubital fossa is reserved for vascular access in life-threatening situations because the vessels are highly visible in this area [17]. Inappropriate sites for PIVC insertion may have been selected due to the unavailability of vessels in more appropriate locations due to age, comorbidities, and high rates of previous hospitalizations and cannulations. In addition, most of the patients were admitted due to an emergency, which may explain why certain cannulation sites were chosen. On the other hand, during the two-year pandemic period, there was no validated procedure or training regarding the insertion of the peripheral cannula.

Of the 3377 catheters, only 1030 (31.1%) were removed due to the planned termination of IV therapy. Most of them stopped functioning due to pathological reasons. Phlebitis is reported to be the most frequent and highly prevalent reason for the complications related to PIVCs. A recent meta-analysis showed that the pooled proportion of phlebitis was 19.3% [18]. In our study, the rate of catheter removal due to phlebitis was lower than that in other studies, with only 7.3% being proven cases. However, based on the assessment of the cannulation site using the Phlebitis Scale, possible first symptoms of vasculitis occurred in 10.5% of the cases. Nevertheless, according to the Infusion Nursing Society, the accepted phlebitis rate is 5% and less [19]. Determining the reasons for such a high percentage of phlebitis in the studied population may be difficult because the literature data indicate a multifactorial basis for this complication, starting from the choice of the cannulation site, through the aseptic procedure of its insertion, and to the control of the already functioning vascular line [19]. The occurrences of leakage (26.6%) and occlusion (17.5%) were significantly higher than in the meta-analysis that was already cited when such complications appeared, with a rate of 8% for occlusion and 7.3% for leakage. There was a significant correlation between suboptimal device locations and the most frequent complications. In the multivariate analysis performed by Wallis et al., it was found that the modifiable risk factors for occlusion as well as accidental removal included PIVC location, especially in the hand or antecubital fossa [20]. Additionally, patients removing PIVCs by themselves is a significant problem (13%). In the current IT and VAD procedures, great attention is paid to take care of the VAD and to achieve the early diagnosis of venous inflammation. Selecting the right place to insert the catheter and educating the patient on how to avoid accidental removal and prolong the PIVC’s dwell time should be part of the standard procedure according to the guidelines [17]. Despite their inclusion in the protocol, extravasation, thrombosis, and catheter-related bloodstream infections were not detected in the PIVC documentation. It may seem that reduced supervision during the procedure and less training in cannulation and infusion may lead to inadequate recognition of these complications. Another aspect that has a direct impact on the quality of care for hospitalized patients and the proper performance of procedures such as PIVC insertion and maintenance is the number of available qualified medical staff. Poland is among the countries with the lowest numbers of nurses per population in relation to the European average, with 5.7 nurses vs. 8.5 per 1000 people, respectively [21]. The COVID-19 pandemic has placed an additional burden on the healthcare system due to the need to increase the care capacity with limited resources. The sector of medical staff that was the most affected included staff in direct contact with patients, namely nurses and doctors, which could have interfered with the individual elements of patient care, including cannulation [11].

Intravenous therapy is difficult to characterize due to the very large variety of therapeutic agents used. Many drugs do not have established DDDs, which does not allow their consumption rates to be directly compared to others. Among the admission causes for IMDs, chronic heart diseases rank first, especially for older patients [22]. The oral forms of loop diuretics are widely used in the chronic treatment of stable patients with congestive heart failure. During disease exacerbations, tissue swelling occurs, which may impair the absorption of drugs from the gastrointestinal tract; thus, intravenous therapy is needed to rehydrate patients, especially during severe manifestations of cardiovascular diseases, such as pulmonary oedema [23].

Glucocorticoids (GCs) are used to treat diverse clinical conditions due to their pleiotropic effects. GCs’ good absorption from the gastrointestinal tract allows them to be successfully used in oral form, except in situations where the clinical condition of the patient precludes oral administration. Considering this, using GCs in their parenteral form seems to be unreasonable. On the other hand, in the examined ward, most of the patients were admitted for urgent reasons, which, due to their serious general condition, could have been the reason for choosing this form of intravenous glucocorticosteroid administration. However, further research is needed to assess the appropriateness of their current application. 

Antimicrobials were the most important group of drugs in our study, with the total consumption being 600 DDDs/1000 pds, which was lower than the value obtained from the meta-analysis conducted by Bitterman et al. for hospital departments of internal medicine, which was 677 DDDs/1000 pds (95% confidence interval: 634–720) [24]. Nevertheless, 0.6 DDDs of antibiotics per day demonstrate that more than half of the patients received at least one antibiotic per day during their stay in the studied ward, while the infections were the reason why only 20% of the patients were admitted, including virus infections. During the COVID-19 pandemic, three-quarters of the patients received antibiotics, with a much lower percentage of bacterial co-infections [25]. The heterogeneous course of this new disease, the initial lack of scientific evidence, and experience with bacterial co-infections in pneumonia caused by influenza virus may be the reasons for such abuse of this group of drugs [26,27]. Notably, the misuse of antibiotics in patients hospitalized due to COVID-19 without a confirmed bacterial co-infection could have influenced the bias in the quality and quantity of antibiotics used during the analyzed pandemic period. On the other hand, it should be noted that our analysis concerned the final diagnosis, and, during their stay, the patients may have suffered from infections and required antibiotics, which could also have influenced the increase in their consumption. The appropriate treatment of infections is dependent on a functioning vascular line because most antibiotics administered in hospitals do not have an oral form. The consumption of individual groups is dominated by beta-lactam antibiotics (3996 DDDs, 78%), which are time-dependent killers. Their pharmacodynamics are based on the time when their concentrations are above the minimum inhibitory concentration; therefore, they must be provided in appropriate doses at regular intervals. The loss of a PIVC may affect the agent’s serum level and prolong the hospital stay due to the infusion being delayed, resulting in the infection reappearance [28]. 

Dehydration among geriatric patients is known to be related to a higher risk of acute coronary events, pneumonia, and thromboembolism. The reduction in liquid intake is associated with a decreased feeling of thirst, as well as the age-related impairment of kidney function. Dementia, dysphagia, and various types of functional impairment significantly affect the oral intake of fluids [29]. In-hospital IV fluid therapy is commonly used to correct the level of hydration and electrolyte disturbances and is an important part of the treatment for cases of sepsis, acute renal failure, hyponatremia, hypercalcemia, and acute pancreatitis. It is therefore not surprising that fluid therapy has such a large share among the pharmacotherapy types. However, the direct supply of fluids to the bloodstream is a non-physiological method, requires the maintenance of peripheral vascular access, and can also lead to tissue conduction and swelling. Treating fluid deficiency orally should play a primary role whenever possible due to the absence of significant complications [30]. 

The present study has certain limitations. Due to its retrospective nature, it was not possible to precisely determine the communication between the medical staff and the patient, and it was also impossible to directly validate the procedures used. For the same reason, it was not possible to determine the reasons for the staff’s choice of cannulation site. In the case of some medications, no unified methods of consumption were established, which did not enable the precise description and comparison of their consumption.

## 4. Methods

This study was conducted in the internal medicine ward of a regional specialist hospital, the Centre of Pulmonology and Thoracic Surgery in Bystra, Silesian Voivodeship, Poland, from 1 January 2021 to 31 December 2022. Taking into account the expected level of use of PIVC min. 0.6, at least 120 patients must be included to obtain relevant statistical sample size; considering the bias margin, we decided to conduct a 2-year retrospective observational study [https://sample-size.net/sample-size-proportions/, accessed on 3 April 2024]. The inclusion criterion was the establishment and use of peripheral vascular access upon admission or during hospitalization. The exclusion criteria included admission with a PIVC that was implemented in another hospital and the transfer of the patient to another facility prior to the treatment’s completion. In the 2021–2022 period, the studied ward was twice transformed into a temporary unit dedicated only to COVID-19 patients.

The patients’ data (sex, age, main reason for hospitalization (ICD-10), admission mode, length of hospitalization, duration of intravenous therapy, and information on previous hospital stays up to three months prior) were obtained from electronic medical records. The end of hospitalization was classified as a patient’s discharge from the hospital or death. The main reasons for hospitalization based on ICD-10 were divided into two main groups: chronic diseases and infections. The chronic diseases group was divided into 7 subgroups: cancer diseases (excluding patients in the acute phase of the disease); cardiovascular diseases; diseases of the gastrointestinal tract, liver, biliary tract, or pancreas (later described as gastrointestinal diseases); metabolic disorders (including diabetes and lipid and electrolyte disorders); nephrological diseases; respiratory diseases; and patients with anemia, poisoning, pressure ulcers, and musculoskeletal disorders. 

Information was collected on the number of PIVCs and their characteristics, including the location, the reason for its removal, and the number of VADs that failed within 24 h due to pathological reasons. During the studied period, catheters with gauge 20 or 22 were used in the department. The PIVC’s position was classified as optimal or suboptimal based on the Infusion Therapy Standards of Practice, 8th Edition [17]. The optimal location of the PIVC was considered to be within the extensor venous vessels of the upper limbs, i.e., the forearm and upper arm. The flexion areas of the upper limbs, including the wrist and the antecubital fossa, the hand, and the veins of the lower limbs, were considered suboptimal locations. Complications requiring VAD removal are described in accordance with the Infusion Therapy Standards of Practice, 8th Edition, and studies have been carried out in this area [10]. Phlebitis was defined as inflammation of the vessel, which manifested through pain, tenderness, swelling, purulence, or a palpable venous cord. The PIVC evaluation for signs of phlebitis was performed with the 6-step Visual Infusion Phlebitis Scale [31]. Occlusion was defined as the loss of the ability to infuse fluids or medications through a previously functioning catheter. Displacement or removal by the patient meant that the cannula was lost from the site of insertion. Leakage was defined as the outflow of the injected substance from the injection site. Other complications, like a catheter-related bloodstream infection, were not detected. According to intravenous therapy procedure, the indications for the insertion of a peripheral catheter include emergency admission due to severe infections, life-threatening anemia, acute cardiac and pulmonary conditions, electrolyte disorders, dehydration, and acute liver or kidney failure. The procedure for inserting a peripheral intravenous catheter, validated at the time of the research, is described in the Appendix A. There was no measurement of compliance for peripheral catheter insertion and care procedures during the study period. The information about central intravenous catheters was also selected, along with the reason for their insertion. The number of patient days and duration of utilizing a PIVC were calculated based on the CDC guidelines [32]. The PIVC utilization ratio was calculated as the number of PIVC pds (pds, patient days) per the total number of pds. 

The intravenous drugs and preparation methods used in the IMD were classified according to the Anatomical Therapeutic Chemical (ATC) system [33]. The quantitative consumption of medications, for antibiotics and other drugs excluding infusion fluids, was calculated using the defined daily dose (DDD). The DDD is the assumed average maintenance dose per day for a drug used for its main indication in adults and was established by the WHO. The DDD is sometimes a dose that is rarely or never prescribed because it is an average of two or more commonly used doses, so, for the purposes of this study, it was assumed that the average doses provided as the DDD corresponded to the doses used in therapy, which was not consistent with clinical practice but made it possible to perform a comparative analysis. Due to the absence of the DDD, the consumption of infusion fluids and others was quantified in milliliters (mL).

Due to highest consumption rate, antibiotics were analyzed in-depth by considering both routes of administration: intravenous and oral. Individual antimicrobials were presented according to the ATC classifications [33]. In the analyzed group of intravenous antibiotics, the use of cloxacillin, vancomycin, ceftazidime, and cefepime was specified in terms of cases with a documented relationship with catheter failure [8]. Loop diuretics and glucocorticosteroids were also selected from the other groups because of their high consumption rates. Among the therapeutics with undetermined DDDs, electrolyte solutions were selected due to their consumption rate being the highest.

Descriptive statistics, namely the mean and standard deviation or median (Me) and interquartile range (IQR), were calculated for the quantitative variables. The Shapiro–Wilk test was conducted to determine whether they follow a normal distribution. The average values of age, length of stay (LOS), and duration of intravenous therapy (IT) were compared between patients with cannulas that stopped functioning below 24 h and those that functioned for at least 24 h. The *t*-test for normally distributed variables or Cochrane–Cox test for those that do not follow normal distribution was used to make the comparison of average values in groups. The qualitative (categorical) variables were compared using cross-tabulations, and the chi-squared test was used to assess the correlations between them. The *p*-value of 0.05 was used to test significance. Analyses were performed using STATISTICA 10, StatSoft Inc. (Tulsa, OK, USA).

This study was conducted according to the guidelines of the Declaration of Helsinki and approved by the Bioethics Committee of Jagiellonian University (protocol code 1072.6120.4.2023; date of approval 15 February 2023). All data entered in the electronic database and analyzed in this study were anonymized. 

## 5. Conclusions

In the internal medicine department, intravenous therapy with the use of PIVCs remains the main form of treatment. Up to 1/5 of peripheral intravenous catheters are lost within the first 24 h after their insertion, more frequently in patients with infections. Antibiotics constitute the main group of intravenous drugs, which makes proper functioning vascular access an essential component of proper antimicrobial treatment. Additional research on the correct use of intravenous therapy and cannulation procedures is needed to improve the quality of care for patients treated with peripheral intravenous lines.

## Figures and Tables

**Figure 1 antibiotics-13-00664-f001:**
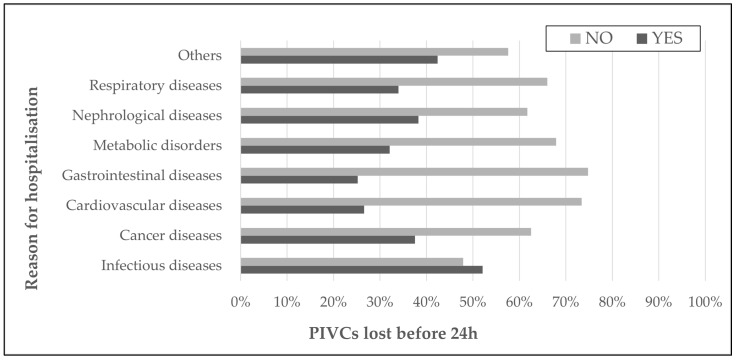
The occurrence of peripheral intravenous catheters (PIVCs) functioning <24 h [%] depending on the reason for hospitalization (*p* < 0.001).

**Figure 2 antibiotics-13-00664-f002:**
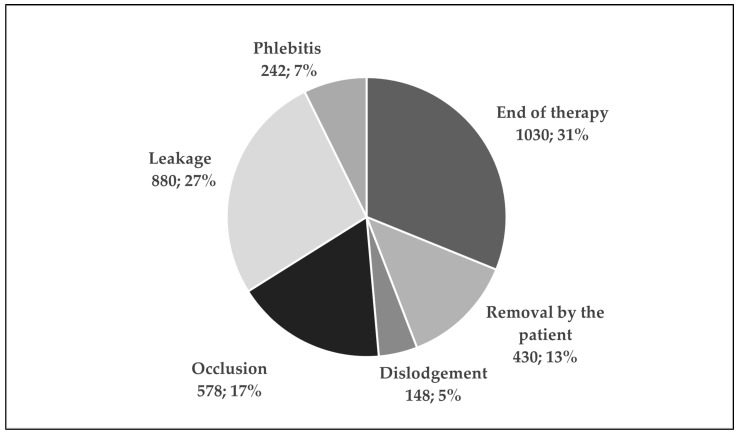
The main reasons for peripheral intravenous catheters removal [n, %].

**Figure 3 antibiotics-13-00664-f003:**
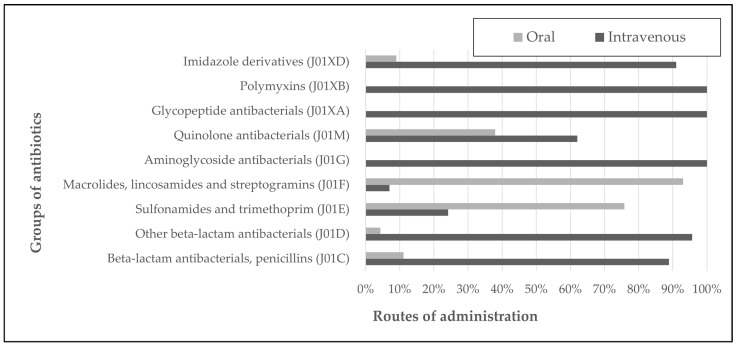
Comparison of total intravenous and oral route of antibiotics consumption in years 2021–2022 [%].

**Table 1 antibiotics-13-00664-t001:** Characteristics of the patients of the studied internal medicine ward in the years 2021–2022.

Study Patient Population			
Total number of admissions (n)	1406		
Patients with PIVC * (n, %)		1176	83.6%
PIVC utilization ratio **		0.8	
Sex (n, %)	Female	578	49.1%
	Male	598	50.9%
Age (years)	Median (IQR)	74 (66–83)	
Hospital admission mode (n, %)	Emergency	1028	87.4%
	Planned	148	12.6%
Main reason for hospitalization (n, %)	Infectious diseases	261	22.0%
	Chronic diseases	915	78.0%
Chronic diseases (n, %)	Cancer diseases	32	3.8%
	Cardiovascular diseases	290	31.7%
	Gastrointestinal tract, liver, biliary tract and pancreas diseases	155	16.9%
	Metabolic disorders	131	14.3%
	Nephrological diseases	79	8.6%
	Respiratory diseases	94	10.3%
	Others	132	14.4%
Admissions due to COVID-19 (n, %)		159	13.5%
Previous hospitalizations up to 3-months back (n, %)		237	20.1%
In-hospital deaths (n, %)		130	11.0%
Fatality case rate (%)	All hospitalized	11.1%	
	Patients with PIVC-	11.2%	
Length of hospital stay (days) (Median (IQR))		9 (6–12)	
Length of intravenous therapy (days) (Median (IQR))		7 (5–11)	

* Peripheral intravenous catheter (PIVC); pds, patient days; ** the PIVC utilization ratio was calculated as the number of PIVC pds per total number of pds.

**Table 2 antibiotics-13-00664-t002:** Characteristics of the peripheral intravenous catheters used in the studied internal medicine ward in the years 2021–2022.

Study Patient Population with Peripheral Intravenous Catheters				
Total number of inserted PIVC (n)		3377		
Number of used PIVC per patient (Me, IQR)		2 (1–4)		
PIVC patient days		9678		
Location of PIVC (n, %)	Optimal	Forearm	955	28.3%
	Suboptimal	Hand	787	23.3%
		Antecubital fossa	598	17.7%
		Wrist	507	15.0%
		Foot	118	3.5%
		Other	409	12.1%
PIVC that stopped functioning within 24 h (n, %)			764	22.6%
Patients with at least one PIVC that stopped functioning within 24 h (n, %)			425	36.1%
Differences in patients’ characteristics		PIVC < 24 h (n); *p* < 0.001		
		Yes	No	
Age [years, Median, (IQR)]		76 (68–84)	73 (64–81)	
Length of stay [days, Median, IQR]		11 (7–15)	8 (6–11)	
Length of intravenous therapy [days, Median, (IQR)]		9 (6–14)	6 (4–9)	
Phlebitis Scale, venous inflammation signs (phlebitis) when removing the PIVC (n, %) *		0—no sign	2617	89.0%
		1—possible 1st signs	309	10.5%
		2—early stage	13	0.4%
		3—medium stage	1	0.1%

PIVC—peripheral intravenous catheter; * excluding catheters removed by the patients themselves as those showing no signs of inflammation.

**Table 3 antibiotics-13-00664-t003:** Intravenous drugs used in the studied internal medicine ward in the years 2021–2022.

ATC Classification System, Only IV Solutions		Consumptions of Drugs *	
		DDD	mL
Anti-infective For Systemic Use	Antibacterials for systemic use (J01)	5009.9	N/A
	Antimycotics for systemic use (J02)	100.0	N/A
	Antivirals for systemic use (J05)	69	N/A
Systemic Hormonal Preparations **, (H)	Corticosteroids for systemic use (H02)	3746.7	N/A
Cardiovascular System (C)	Loop diuretics (C03)	2298.3	N/A
	Others (C01A, C01B, C01CA, C01D *, C02)	728.3	100.0
Alimentary Tract and Metabolism (A)		2195.7	1200.0
Blood And Blood Forming Organs (B)	Electrolyte solutions (B05B *)	N/A	5,705,600.0
	Others (B02, B03A *, B05A *, B05B *)	116.3	620,680.0
Musculoskeletal System (M): M01		133.3	N/A
Nervous System (N): N01B, N02, N05, N05C		1109.2	N/A
Respiratory System (R): R03, R06		9.7	N/A
TOTAL		15,516.4	6,327,580.0
TOTAL per PIVC patient days		1.6	653.8

The Anatomical Therapeutic Chemical (ATC) classification system established by WHO Collaborating Centre for Drug Statistics Methodology. IV—intravenous; * substances with no established defined daily dose (DDD), expressed in milliliters (mL); ** Excl. Sex Hormones and Insulins; N/A—not available or not applicable; PIVC—peripheral intravenous catheter.

**Table 4 antibiotics-13-00664-t004:** Antibiotic consumption, both oral and intravenous solutions in the years 2021–2022.

Antibacterials for Systemic Use (J01)	Route of Administration (DDD)				
	2021		2022		Total Amount
	IV	Oral	IV	Oral	
Beta-lactam antibacterials, penicillins (J01C)	600.0	62.3	915.8	127.0	1705.1
Other beta-lactam antibacterials (J01D)	1139.6	90.0	1340.2	21.0	2590.8
Sulfonamides and trimethoprim (J01E)	72.0	160.0	65.6	272.0	569.6
Macrolides, lincosamides and streptogramins (J01F)	13.7	261.0	27.7	294.0	596.4
Aminoglycoside antibacterials (J01G)	36.5	0.0	84.8	0.0	121.3
Quinolone antibacterials (J01M)	83.8	45.0	60.0	44.5	233.3
Glycopeptide antibacterials (J01XA)	165.5	0.0	160.0	0.0	325.5
Polymyxins (J01XB)	12.0	0.0	6.0	0.0	18.0
Imidazole derivatives (J01XD)	159.7	13.3	67.0	10.0	250.0
SUM	2282.8	631.6	2727.1	768.5	6410.0
TOTAL antibiotic (oral and intravenous) consumption per patient days					0.6

The Anatomical Therapeutic Chemical (ATC) classification system established by WHO Collaborating Centre for Drug Statistics Methodology. DDD—defined daily doses; IV, intravenous.

## Data Availability

The datasets generated during and/or analyzed during the current study are not publicly available but are available from the corresponding author upon reasonable request.

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
