# Peer review of "Peripheral Intravenous Therapy in Internal Medicine Department—Antibiotics and Other Drugs’ Consumption and Characteristics of Vascular Access Devices in 2-Year Observation Study"

_antibiotics, 2024, doi:10.3390/antibiotics13070664_

Round 1

Reviewer 1 Report

Comments and Suggestions for Authors

The authors conducted an observational study on peripheral intravenous therapy in the internal medicine department. Though, there are some limitations regarding the retrospective nature, the results from this study can help identify common practices and variations in care. All the information is constructively detailed. 

Author Response

Reviewer #1, Round 1:
The authors conducted an observational study on peripheral intravenous therapy in the internal medicine department. Though there are some limitations regarding the retrospective nature, the results from this study can help identify common practices and variations in care. All the information is constructively detailed.

Authors’ reply: Thank you for your review and appreciation of our work!

Reviewer 2 Report

Comments and Suggestions for Authors

I have read with interest the manuscript submitted by Piekiełko et al. I have a few comments to be addressed in order to improve the quality of the manuscript:

- all the abbreviations used in the abstract should be described at first use

Table 2 should be rearranged or maybe split into 2 separate tables.

All tables should include a legend describing all abbreviations used.

Table 4 - why just some drugs have the mention "antibacterial"? they are all antibiotics..

Can the authors link the unfavorable outcomes (such as death or longer hospital stay) with the presence of an PIVC or other factors? 

Some more elaborate statistical methods should be inserted in order to properly stratify the scientific soundness of your findings. 

The discussion should start with some general information, then your findings. Moreover, further comparisons with findings from other studies should be performed.

the reference list is scarce and not edited according to the mdpi pattern.

the conclusions should be rephrased and include the findings of your study.

Best regards,

Comments on the Quality of English Language

moderate to extensive English editing and rephrasing is needed.

Author Response

Reviewer #3, Round 1:

I have read with interest the manuscript submitted by Piekiełko et al.

Authors’ reply: Thank you for your review and all suggestions.

All the abbreviations used in the abstract should be described at first use.

Authors’ reply: The abstract has been corrected and all abbreviations have been expanded:

“Peripheral intravenous catheters (PIVCs); Internal Medicine Department (IMD); defined daily doses (DDD)”

Table 2 should be rearranged or maybe split into 2 separate tables.

Authors’ reply: Thank you for this comment. Following your king suggestion, we decided to take a closer look at table 2 to modify it in order to present our results more clearly. A comparison of the occurrence of intravenous lines functioning for less than 24 hours depending on the reason for hospitalization from Table 2 is now presented in Figure 1. The main reasons for PIVC removal were presented in Figure 2. The manuscript was supplemented with figures due to other reviewer suggestion. 

All tables should include a legend describing all abbreviations used.

Authors’ reply: Thank you, all tables have been supplemented with appropriate legends.

Table 4 – why just some drugs have the mention “antibacterial”? they are all antibiotics…

Authors’ reply: We agree with the reviewer's opinion that all presented drugs are antibiotics but in our study, we used The Anatomical Therapeutic Chemical (ATC) classification system established by WHO Collaborating Centre for Drug Statistics Methodology and that is why this nomenclature of WHO`s is used in the table 4.

Can the authors link the unfavorable outcomes (such as death or longer hospital stay) with the presence of an PIVC or another factor?

Authors’ reply: We consider your comment valuable for improving the quality of our results. We decided to compare deaths among the entire patient population in relation to deaths among patients with peripheral catheters as the fatality case rate. Due to  changes section “Results” were improved. Table 1 was also supplemented with fatality case rate, as below:

Results

” … The number of deaths in the entire population hospitalized during studied period was 158, while among patients with PIVC was 130. The PIVC-fatality case rate was 11.1%, similar compared to the total fatality case rate which was 11.2% (Table 1).”

Some more elaborate statistical methods should be inserted in order to properly stratify the scientific soundness of your findings.

Authors’ reply: Due to your kind suggestion, the section “Methods” in part considering statistical analysis was corrected, as below:

“…The Shapiro-Wilk test was conducted to determine whether they follow a normal distribution.”

“…The T-test for normally distributed variables or Cochrane–Cox test for those which do not follow normal distribution was used to make the comparison of average values in groups.”

The discussion should start with some general information, then your findings. Moreover further comparisons with findings from other studies should be performed.

Authors’ reply: According to your review we modified section “Discussion” with some general information and also, we added some new comparisons with other research, as below:

“In the studied department, patients with PIVC constituted 83.6% of all hospitalized and almost ¼ of peripheral catheters stopped functioning before 24 hours. The length of IT therapy was 7 days compared to 9 days of hospital stay length. Most of the PIVCs were located in suboptimal places. Among peripheral intravenous therapy, antibiotics were the dominant group of administered drugs. Furthermore, in total antibiotics consumption, the IV route was the main one.”

And:

“... Nevertheless, according to Infusion Nursing Society, the accepted phlebitis rate is 5% and less [19]. Determining the reasons for such a high percentage of phlebitis in studied population may be difficult, because literature data indicate a multifactorial basis for this complication, starting from the choice of the cannulation site, through the aseptic procedure of its insertion, to the control of the already functioning vascular line [19].”

The reference list is scarce and not edited according to the MDPI pattern.

Authors’ reply: Thank you. We took a closer look at all the references and corrected them due to MDPI recommendations.

The conclusions should be rephrased and include the findings of your study.

Authors’ reply: The authors greatly appreciate your comments regarding the conclusions. The section was edited as below:

“In the Internal medicine department, intravenous therapy with the use of PIVCs remains the main form of treatment. Up to 1/5 of peripheral intravenous catheters are lost within the first 24 hours after their insertion, more frequent in patients with infections. Antibiotics constitute the main group of intravenous drugs, which makes proper functioning vascular access an essential component of proper antimicrobial treatment. Ad-dictional research on the correct use of intravenous therapy and cannulation procedures is needed to improve the quality of care for patients treated with peripheral intravenous lines.”

According to changes in conclusion section, the abstract was also edited, as below:

“…Up to 1/5 of peripheral intravenous catheters are lost within the first 24 hours after their insertion with most of them placed non-optimally. A properly functioning PIVC appears to be crucial for antimicrobial treatment.”

Reviewer 3 Report

Comments and Suggestions for Authors

General comments

This is a two-year retrospective single-centred observation study conducted in Poland that looked at the use of PIVC and its association with complications.

Overall, the manuscript is generally well written but requires some minor modifications for clarity (see specific comments below). However, the study looks more like a descriptive study that readers may not be able to easily grasp the main message and big pictures. For instance, the PIVC reason for removal and lack of optimal locations available are old news rather than anything new in the scientific community. The need of PIVC for intravenous antibiotics is a well-known fact. Besides, it seems like the focus of the article is about PIVC rather than antibiotics. The defined daily doses (DDD) are for antibiotics but not for other classes of drugs. This makes me wonder whether this MDPI Antibiotic journal is the best place for publication of this manuscript.

Some of the authors’ claims in the discussion require more justifications (See specific comments).

The conclusion in the abstract that state “PIVC and IT procedures are the most important ones requiring constant supervision and training” require more evidence and justification, as the current study did not demonstrate how supervision and training improved outcomes.

Specific comments

Abstract

Line 15-28: As a general rule of thumb, all abbreviations not commonly used must be defined. It could be impossible for readers to understand what PIVC, IMD, DDD are without reading the content of the manuscript.

Introduction

Line 46-48: It seems like there are two contradictory sentences here. The first sentence described nursing staff as the IT supervisors; the second sentence described the success of IT is dependent on the patient and medical staff. Then, who and what is the main factor per the authors’ opinions? Or are they saying this is multi-factorial? Can some references be provided to support their claim?

Line 55-56: The part about “antibiotics associated with increased PIVC failure through their potentially irritating effects on blood vessels” may need to be more specific. Not all antibiotics can cause phlebitis.

Line 64-65: The part about the challenge of providing VAD care and maintaining proper supervision may require some evidence or references to support.

Methods

Line 70-142: What is the minimum sample size required, determined prior to the study?

Results

Line 163: Was the “52.1% vs. 47.9%” statistically significant? Please include the p-value.

Line 168: It is good that the table included reasons for PIVC removal. But a more important question here is the reasons for PIVC insertions and whether they are warranted with good clinical indications. It is because it seems like the discussion paragraphs are critiquing overuse of PIVC (e.g. glucocorticosteroids, hydration) without showing data in the current study whether these PIVC insertion are all justifiable or not.

Line 174: “the location of the PIVC in a suboptimal place and the occurrence of complication” -this needs to be better clarified. Do you mean “The flexion areas of the upper limbs, including the 95 wrist and the antecubital fossa, the hand and the veins” as described in Line 95-96. Or simply refer to Table 2.

Discussion

Line 199: “indicate the need to use as many as two for a treatment of this duration” This sentence sounds a bit confusing. Do you mean as many as two PIVC for an IT of 7 days?

Line 207-208: Not all antibiotics causing vascular damage and irritation in the same degree. I see that Table 4 lists the group of antibiotics. In order to support your argument that the PIVC failure in the study are due to antibiotics causing vascular damage and irritation, perhaps, you should show us certain antibiotics known to cause phlebitis (e.g. cloxacillin) had a higher likelihood of PIVC failure.

Line 246: The part about “Selecting the right place to insert the catheter” – are we certain that the suboptimal placement site of PIVC is due to clinicians not knowing where the right site are? Or could be it be due to difficulty in using the optimal site, as described in Line 223-224?

Line 273: Since you stated using glucorticosteroids in parenteral form seems unreasonable, did you look into your study to determine the reasons why they are used? Or else this paragraph seems to be simply generating a postulation without even attempting to justify the claim.

Line 280-284: You brought up a good point that half of the patients received antibiotics but only 20% of the patients had diagnosed infections. But it is also important to acknowledge the recorded diagnosis here can be based on a discharge diagnosis after most of the investigation results are back. In an acute setting, most clinicians would need to follow the sepsis protocol that require the use of antimicrobials, even when the diagnosis is not yet confirmed. Another thing of note is patients could develop any secondary illness during their hospital stay, such as urinary tract infections and hospital-acquired pneumonia. Therefore, even though infections are not their main diagnoses for admission, they still have valid reasons requiring antimicrobials.

Line 306-308: Although this paragraph tries to justify the use of oral rehydration therapy, there is no information on whether the IV therapy for dehydration and electrolyte replacement are unjustifiable in the current study. It is unclear how many patients of the current study need the PIVC for dehydration and electrolyte imbalance. I imagine the clinicians would not go through the troubles of inserting a PIVC and running the risk of complications if oral options are indeed available.

Conclusion

Line 318-320: The current study did not seem to have data to support that antibiotics were overused during the pandemic.

Line 323-324: The current study did not seem to have data to support how regular training improved quality of healthcare.

Comments on the Quality of English Language

No major errors detected except more clarifications are needed for the abbreviations used.

Author Response

Reviewer #4, Round 1:

This is a two-year retrospective single-centered observation study conducted in Poland that looked at the use of PIVC and its association with complications.

Overall, the manuscript is generally well written but requires some minor modifications for clarity (see specific comments below). However, the study looks more like a descriptive study that readers may not be able to easily grasp the main message and big pictures. For instance, the PIVC reason for removal and lack of optimal locations available are old news rather than anything new in the scientific community. The need of PIVC for intravenous antibiotics is a well-known fact. Besides, it seems like the focus of the article is about PIVC rather than antibiotics. The defined daily doses (DDD) are for antibiotics but not for other classes of drugs. This makes me wonder whether this MDPI Antibiotic journal is the best place for publication of this manuscript.

Authors’ reply: The authors greatly appreciate your comments regarding our manuscript. We found all your suggestions significant for better presenting the results of our work.

In estimating antibiotic consumption, we used The Anatomical Therapeutic Chemical (ATC) classification system established by WHO Collaborating Centre for Drug Statistics Methodology. All defined daily doses for individual drugs, not only for antibiotics were prepared due to the mentioned source. 

The conclusion in the abstract that state “PIVC and IT procedures are the most important ones requiring constant supervision and training” require more evidence and justification, as the current study did not demonstrate how supervision and training improved outcomes.

Authors’ reply: We very much agree with your suggestion! Due to the lack of intervention, the presented study does not show the role of proper supervision in improving outcomes. Further research is necessary to clarify this issue. After careful consideration, we modified the entire conclusions section, which was also included in the abstract, as below:

Conslusions:

“In the Internal medicine department, intravenous therapy with the use of PIVCs remains the main form of treatment. Up to 1/5 of peripheral intravenous catheters are lost within the first 24 hours after their insertion, more frequent in patients with infections. Antibiotics constitute the main group of intravenous drugs, which makes proper functioning vascular access an essential component of proper antimicrobial treatment. Additional research on the correct use of intravenous therapy and cannulation procedures is needed to improve the quality of care for patients treated with peripheral intravenous lines.”

And abstract:

“…Up to 1/5 of peripheral intravenous catheters are lost within the first 24 hours after their insertion with most of them placed suboptimally. A properly functioning PIVC appears to be crucial for antimicrobial treatment.”

Specific comments

Abstract, Line 15-28: As a general rule of thumb, all abbreviations not commonly used must be defined. It could be impossible for readers to understand what PIVC, IMD, DDD are without reading the content of the manuscript.

Authors’ reply: Again, we fully agree and thank you for your comment! The abstract has been corrected and all abbreviations have been expanded, as below:

“Peripheral intravenous catheters (PIVCs); Internal Medicine Department (IMD); defined daily doses (DDD)”

Introduction, Line 46-48: It seems like there are two contradictory sentences here. The first sentence described nursing staff as the IT supervisors; the second sentence described the success of IT is dependent on the patient and medical staff. Then, who and what is the main factor per the authors’ opinions? Or are they saying this is multi-factorial? Can some references be provided to support their claim?

Authors’ reply: As you suggested, we took a closer look at those lines and decided to clarify them. Nursing staff are the IT supervisors but the success of the whole IT procedure including PIVC insertion and maintenance depends not only on the nurses, but also on the patient himself and factors related to him. To support our claim, we extended this paragraph (as below) and provided some references:

“In a medical treatment team, the nursing staff are usually the IT supervisors. The success of IT using a PIVC is dependent on many factors related to the patient clinical status and behaviour and the nursing staff decisions. It is the responsibility of the nursing staff to choose the appropriate place for VAD insertion and to take care of the vascular line. The patient's responsibility is to be aware of having a catheter, its limitations and reporting disturbing symptoms that may indicate its malfunction.”

Line 55-56: The part about “antibiotics associated with increased PIVC failure through their potentially irritating effects on blood vessels” may need to be more specific. Not all antibiotics can cause phlebitis.

Authors’ reply: According to the review, the paragraph has been supplemented with specific examples of drugs according to data from the literature, as below:

”Antibiotics, despite their undeniable role in treating infections, are also associated with increased PIVC failure through potentially irritating effects on blood vessels for some of them such as flucloxacillin, vancomycin, ceftazidime and cefepime [8].”

Line 64-65: The part about the challenge of providing VAD care and maintaining proper supervision may require some evidence or references to support.

Authors’ reply: Thank you for your comment. References supporting paragraph were added.

Methods, Line 70-142: What is the minimum sample size required, determined prior to the study?

Authors’ reply: Considering PIVC utilization ratio min. 0.6, at least 120 patients must be included to get relevant statistical sample size, taking bias margin we decided to conduct a 2-year retrospective observational study (https://sample-size.net/sample-size-proportions/). During analyzed period (2021-2022) 1,406 patients were hospitalized in the Internal Medicine Department and those required PIVC constituted 83.6% (n=1,176). We believe that the group of patients included in the study is representative of the hospitalized population in the department. The “Methods” section was supplemented.

Results, Line 163: Was the “52.1% vs. 47.9%” statistically significant? Please include the p-value.

Authors’ reply: Corrected, the p value is below 0.001 and was included in the manuscript:

”The patients hospitalized due to infections were more likely to have at least one PIVC that functioned for <24h, respectively, 52.1% vs. 47.9%; p<0.001 (Table 2).”

Line 168: It is good that the table included reasons for PIVC removal. But a more important question here is the reasons for PIVC insertions and whether they are warranted with good clinical indications. It is because it seems like the discussion paragraphs are critiquing overuse of PIVC (e.g. glucocorticosteroids, hydration) without showing data in the current study whether these PIVC insertion are all justifiable or not.

Authors’ reply: Thank you for your comment. We agree that PIVC removal and insertion reasons are both clinically important. Our study data was obtained retrospectively from medical records which provided detailed information only about PIVC removals. Due to your kind suggestion we decided to complement our “Methods” section with clinical indication for PIVC insertion from the document describing the procedure, as below:

“…According to intravenous therapy procedure, the indications for the insertion of a peripheral catheter include emergency admission due to severe infections, life-threatening anemia acute cardiac and pulmonary conditions, electrolyte disorders, dehydration, and acute liver or kidney failure.”

Line 174: “the location of the PIVC in a suboptimal place and the occurrence of complication” -this needs to be better clarified. Do you mean “The flexion areas of the upper limbs, including the 95 wrist and the antecubital fossa, the hand and the veins” as described in Line 95-96. Or simply refer to Table 2.

Authors’ reply: Suboptimal places were defined as the flexion areas of the upper limbs, including wrist, antecubital fossa, the hand and the veins of the lower limbs. To correct this sentence, due to your kind suggestion, we refer to table 2.

Discussion, Line 199: “indicate the need to use as many as two for a treatment of this duration” This sentence sounds a bit confusing. Do you mean as many as two PIVC for an IT of 7 days?

Authors’ reply: The sentence has been reworded and we hope the message is now clear, as below:

”According to recommendations, a single PIVC should provide vascular access for approximately 7 days of IT, but the results indicate the need to use as many as two peripheral catheters for a treatment of this duration [4,15]”.

Line 207-208: Not all antibiotics causing vascular damage and irritation in the same degree. I see that Table 4 lists the group of antibiotics. In order to support your argument that the PIVC failure in the study are due to antibiotics causing vascular damage and irritation, perhaps, you should show us certain antibiotics known to cause phlebitis (e.g. cloxacillin) had a higher likelihood of PIVC failure.

Authors’ reply: Due to your kind suggestion we decided to underline those antibiotic agents which have potential to damage blood vessels and may lead to catheter loss. We supplemented section “Methods”, “Results” and “Discussion”, as below:

Methods

“In the analyzed group of intravenous antibiotics, the use of cloxacillin, vancomycin, ceftazidime and cefepime was specified as those with a documented relationship with catheter failure.”

And Results:

“…Antibiotic agents likely to be associated with catheter failure among all intravenous antibiotics, constituted 21.8% (1091.5 DDD). Cloxacillin was the majority with 753.5 DDD followed by vancomycin with 325.5 DDD. Ceftazidime consumption was 12.5 DDD and cefepime was not used.”

The Discussion section was also rephrased to:

“…The use of antibiotics related to the development of complications such as vasculitis accounted for as much as 1/5 of total consumption, which could also have a significant impact on the results [8].”

Line 246: The part about “Selecting the right place to insert the catheter” – are we certain that the suboptimal placement site of PIVC is due to clinicians not knowing where the right site are? Or could be it be due to difficulty in using the optimal site, as described in Line 223-224?

Authors’ reply: The authors believe that clinicians are aware of the optimal sites for PIVC insertion. We tried to estimate the reasons why most lines were placed in suboptimal locations as described in lines 223-224. However, the retrospective nature of the study does not allow for a direct assessment of the reason for choosing a specific site for cannula insertion. The section “Discussion” in part about study limitations was supplemented as below:

“…For the same reason, it was not possible to determine the reasons for the staff's choice of cannulation site.”

Line 273: Since you stated using glucorticosteroids in parenteral form seems unreasonable, did you look into your study to determine the reasons why they are used? Or else this paragraph seems to be simply generating a postulation without even attempting to justify the claim.

Authors’ reply: Thank you for this comment. In our study we found that 87.9% of patients were admitted for emergency indications, which, due to their severe general condition, could have been the reason for choosing the intravenous form of glucocorticosteroid administration.

“…On the other hand, in the studied department, most patients were admitted for urgent reasons, which, due to their serious general condition, could have been the reason for choosing this form of intravenous glucocorticosteroid administration.”

Line 280-284: You brought up a good point that half of the patients received antibiotics but only 20% of the patients had diagnosed infections. But it is also important to acknowledge the recorded diagnosis here can be based on a discharge diagnosis after most of the investigation results are back. In an acute setting, most clinicians would need to follow the sepsis protocol that require the use of antimicrobials, even when the diagnosis is not yet confirmed. Another thing of note is patients could develop any secondary illness during their hospital stay, such as urinary tract infections and hospital-acquired pneumonia. Therefore, even though infections are not their main diagnoses for admission, they still have valid reasons requiring antimicrobials.

Authors’ reply: As you rightly noted, our analysis concerned only the final diagnosis and some patients may have required antibiotics during their stay due to a potential infection. The discussion was supplemented, as below:

“…On the other hand, It should be noted that our analysis concerned the final diagnosis, and during their stay, patients may have suffered from infections and required antibiotics, which could also have influenced the increase in their consumption.”

Line 306-308: Although this paragraph tries to justify the use of oral rehydration therapy, there is no information on whether the IV therapy for dehydration and electrolyte replacement are unjustifiable in the current study. It is unclear how many patients of the current study need the PIVC for dehydration and electrolyte imbalance. I imagine the clinicians would not go through the troubles of inserting a PIVC and running the risk of complications if oral options are indeed available.

Authors’ reply:  Once again, you paid attention to an important issue. We agree that it is very difficult to determine how many patients of our study need the PIVC for only dehydration or electrolyte imbalance because most of them required PIVC for the administration of many different substances during hospital-stay. On the other hand we tried to highlight that proper fluid balance is part of recommendations for action in different clinical conditions. We believe that clinicians use intravenous fluid therapy only when absolutely necessary. We would like to point out that the intravenous route of fluid administration is a nonphysiological route, especially considering the fluid consumption estimated in our study which seems to be high.

 Conclusion, Line 318-320: The current study did not seem to have data to support that antibiotics were overused during the pandemic. Line 323-324: The current study did not seem to have data to support how regular training improved quality of healthcare.

Authors’ reply: Thank you for your comments. After careful consideration, we modified the entire conclusions section, which was also included in the abstract, as below:

Conslusions:

“In the Internal medicine department, intravenous therapy with the use of PIVCs remains the main form of treatement. Up to 1/5 of peripheral intravenous catheters are lost within the first 24 hours after their insertion, more frequent in patients with infections. Antibiotics constitute the main group of intravenous drugs, which makes proper functioning vascular access an essential component of proper antimicrobial treatment. Additional research on the correct use of intravenous therapy and cannulation procedures is needed to improve the quality of care for patients treated with peripheral intravenous lines.”

Reviewer 4 Report

Comments and Suggestions for Authors

Piotr Piekiełko et al present an original article on intravenous therapy in an internal medicine department with a focus on drug consumption und vascular access device. 

This article is potentially interesting, however there are some issues the authors should urgently address: 

- the introduction is too superficial and should be written in a more scientific way

- from my point of view two years of analysis are to short, especially due to COVID-19. I would recommend to include at least one more year. 

- antibiotics should be presented in more details and for each year separately - were there any changes

- how many patients had  a central line?

- which size of PIVCs was used?

- the discussion is also to superficial

- I would highly recommend to include at least 3 figures

To sum up, this is a very important study, which urgently needs major revisions. I'm looking forward to receive the revised version of the manuscript as soon as possible, as it adresses a highly relevant topic for scientist and physicians worldwide. 

Comments on the Quality of English Language

Extensive editing of English language required

Author Response

Reviewer #5, Round 1:

Piotr Piekiełko et al present an original article on intravenous therapy in an internal medicine with a focus on drug consumption and vascular access device. This article is potentially interesting, however there are some issues the authors should urgently address:

Authors’ reply: Thank you for finding our manuscript interesting and for helping us properly present the results of our study.

The introduction is too superficial and should be written in a more specific way

Authors’ reply: Due to your suggestion we have made some changes to the introduction and added appropriate references, as below:

“The success of IT using a PIVC is dependent on many factors related to the patient clinical status and behavior and the nursing staff decisions. It is the responsibility of the nursing staff to choose the appropriate place for VAD insertion and to take care of the vascular line. The patient's responsibility is to be aware of having a catheter, its limitations and reporting disturbing symptoms that may indicate its malfunction. [5]”

“…Antibiotics, despite their undeniable role in treating infections, are also associated with increased PIVC failure through their potentially irritating effects on blood vessels for some of them such as cloxacillin, vancomycin, ceftazidime and cefepime [8].”

From my point of view two years of analysis is too short, especially due to COVID-19. I would recommend to include at least one more year.

Authors’ reply: We appreciate your recommendation and agree that other years data would be valuable to our work. Unfortunately, from 2023, the profile of the department was changed to focus on cardiological diseases, which significantly influenced the type of services provided. We are afraid that including patient data after this change could have a negative impact and distort especially the consumption of antibiotics, which is why we limited the scope of the collected information to 2021-2022. On the other hand, fortunately, we calculated the minimum sample size before starting our research – the “Methods” section was supplemented, as:

“Taking into account the expected level of use of PIVC min. 0.6, at least 120 patients must be included to get relevant statistical sample size, taking bias margin we decided to conduct a 2-year retrospective observational study [https://sample-size.net/sample-size-proportions/].”

Antibiotics should be presented in more details and for each year separately – were there any changes

Authors’ reply: Due to your kind suggestion we have made appropriate calculations and redesigned Table 4 to show antibiotic consumption for both years separately. We have recalculated and corrected some values. The changes were described also in section “Results”, as below:

“…In 2022, compared to 2021, there was an increase in the consumption of IV beta-lactams (2256 DDD vs. 1739.6 DDD) and aminoglycosides (84.8 DDD vs. 36.5 DDD), and a de-crease in the consumption of imidazole derivatives (67 DDD vs, 159.7 DDD). In oral administration, an increase in penicillin (127 DDD vs. 62.3 DDD), sulfonamides and trimethoprim (272 DDD vs. 160 DDD) was noted, and a decrease in other antibiotics from the beta-lactam group (21 DDD vs. 90 DDD), (Table 4).”

Also, due to other reviewer comment we highlighted antibiotics which have potential to damage blood vessels and may lead to catheter loss – cloxacillin, vancomycin, ceftazidime, cefepime. The sections “Methods”, “Results” and “Discussion” were supplemented, as below:

Methods

“In the analyzed group of intravenous antibiotics, the use of cloxacillin, vancomycin, ceftazidime and cefepime was specified as those with a documented relationship with catheter failure.”

And Results:

“…Antibiotic agents likely to be associated with catheter failure among all intravenous antibiotics, constituted 21.8% (1091.5 DDD). Cloxacillin was the majority with 753.5 DDD followed by vancomycin with 325.5 DDD. Ceftazidime consumption was 12.5 DDD and cefepime was not used.”

The Discussion section was also rephrased to:

“…The use of antibiotics related to the development of complications such as vasculitis accounted for as much as 1/5 of total consumption, which could also have a significant impact on the results [7].”

How many patients has a central line?

Authors’ reply: 16 patients had central line due to difficulties with PIVC insertion. Appropriate information were added to sections “Methods” and “Results”, as below:

Methods

“…The information about central intravenous catheters was also selected along with the reason for their insertion.”

Results

“…During the study period, 16 central venous catheters were placed, all because PIVC placement was not possible.”

Which size of PIVCs was used?

Authors’ reply: In the studied unit, PIVC gauge 20 or 22 was used. The section “Methods” was supplemented, as below:

“…During the period under study, catheters with gauge 20 or 22 were used in the department.”

The discussion is also too superficial.

Authors’ reply: The section “Discussion” was supplemented with some new threads regarding reviewers’ suggestions, as below:

“In the department studied, patients with PIVC constituted 83.6% of all hospitalized and 22.6% of peripheral catheters stopped functioning before 24 hours. The length of IT therapy was 7 days compared to 9 days of hospital stay length. Most of the PIVCs were located in suboptimal places. Among peripheral intravenous therapy, antibiotics were the dominant group of administered drugs. Furthermore, in total antibiotics consumption, the IV route was the main one.”

And:

“...Nevertheless, according to Infusion Nursing Society, the accepted phlebitis rate is 5% and less [19]. Determining the reasons for such a high percentage of phlebitis in studied population may be difficult, because literature data indicate a multifactorial basis for this complication, starting from the choice of the cannulation site, through the aseptic procedure of its insertion, to the control of the already functioning vascular line [19].”

And:

“…The use of antibiotics related to the development of complications such as vasculitis accounted for as much as 1/5 of total consumption, which could also have a significant impact on the results [8].”

And:

“…On the other hand, in the examined ward, most patients were admitted for urgent reasons, which, due to their serious general condition, could have been the reason for choosing this form of intravenous glucocorticosteroid administration.”

And:

“…On the other hand, it should be noted that our analysis concerned the final diagnosis, and during their stay, patients may have suffered from infections and required antibiotics, which could also have influenced the increase in their consumption.”

And:

“…For the same reason, it was not possible to determine the reasons for the staff's choice of cannulation site.”

I would highly recommend to include at least 3 figures.

Authors’ reply: Following your kind suggestion we prepared some figures.

    • Fig 1. shows the occurrence of PIVCs functioning <24 hours depending on the reason for hospitalization.
    • 2 presents the main reasons for PIVC removal.
    • Fig 3. Comparison of total intravenous and oral route of antibiotics consumption in years 2021-2022 (%)
    • Additionally, we prepared the graphical abstract.

To sum up, this is a very important study, which urgently needs major revisions. I’m looking forward to receive the revised version of the manuscript as soon as possible, as it addresses a highly relevant topic for scientists and physicians worldwide.

Authors’ reply: The authors greatly appreciate your comments regarding our work. We hope that the revised version of the manuscript in accordance with the reviewers' suggestions will gain your approval.

Round 2

Reviewer 2 Report

Comments and Suggestions for Authors

I appreciate the author's efforts in addressing my comments. The quality of the manuscript has significantly improved. My only remark would be that the newly inserted figures should have an enhanced contrast.

Best regards,

Comments on the Quality of English Language

fine

Reviewer 3 Report

Comments and Suggestions for Authors

The authors have addressed most of my concerns. I do not have anything to add.

Comments on the Quality of English Language

No major errors detected.

Reviewer 4 Report

Comments and Suggestions for Authors

The authors have significantly improved the manuscript. Congratulations. From my point of view the research is of high relevance and should be published as soon as possible.